# Newborn Screening Long-Term Follow-Up Clinics (Continuity Clinics) in the Philippines during the COVID-19 Pandemic: Continuing Quality Patient Care

**DOI:** 10.3390/ijns9010002

**Published:** 2022-12-29

**Authors:** Ebner Bon G. Maceda, Michelle E. Abadingo, Karen Asuncion R. Panol, Frederick David E. Beltran, Ivy Rose C. Valdez-Acosta, Grandelee D. Taquiqui, Sharon B. Gawigawen, Maria Victoria L. Macalino, Laura Maria Soledad M. Aguirre-Aguinaldo, Marive A. Flores-Declaro, Karen June V. Ventilacion, Ma. Rita Anna Salve R. Boligao, Nancy G. Honor, Mirasol S. Ellong, Rona D. Ocho-Ortencio, Genelynne J. Beley, Maria Christina N. Bondoc-Eran, Bradford L. Therrell, Carmencita D. Padilla

**Affiliations:** 1Newborn Screening Reference Center, National Institutes of Health, University of the Philippines Manila, Manila 1000, Philippines; 2Institute of Human Genetics, National Institutes of Health, University of the Philippines Manila, Manila 1000, Philippines; 3Philippine General Hospital, University of the Philippines Manila, Manila 1000, Philippines; 4Ilocos Training and Regional Medical Center, San Fernando 2000, Philippines; 5Cagayan Valley Medical Center, Tuguegarao City 3500, Philippines; 6Baguio General Hospital and Medical Center, Baguio City 2600, Philippines; 7Jose B. Lingad Memorial Regional Hospital, San Fernando 2000, Philippines; 8General Emilio Aguinaldo Memorial Hospital, Trece Martires 4109, Philippines; 9Bicol Regional Hospital and Medical Center, Legazpi City 4500, Philippines; 10Department of Pediatrics, West Visayas State University Medical Center, Iloilo City 5000, Philippines; 11Vicente Sotto Memorial Medical Center, Cebu City 6000, Philippines; 12Eastern Visayas Medical Center, Tacloban City 6500, Philippines; 13Zamboanga City Medical Center, Zamboanga City 7000, Philippines; 14Northern Mindanao Medical Center, Cagayan de Oro City 9000, Philippines; 15Southern Philippines Medical Center, Davao City 8000, Philippines; 16Cotabato Regional & Medical Center, Cotabato City 9600, Philippines; 17National Newborn Screening and Global Resource Center, Austin, TX 78759, USA; 18Department of Pediatrics, School of Medicine, University of Texas Health Science Center at San Antonio, San Antonio, TX 78229, USA; 19Department of Pediatrics, College of Medicine, University of the Philippines Manila, Manila 1000, Philippines

**Keywords:** newborn screening, continuity clinic, COVID-19, long-term follow-up

## Abstract

The COVID-19 pandemic has challenged healthcare systems worldwide. In the Philippines, long-term care for patients with conditions identified through newborn screening (NBS) is coordinated through Newborn Screening Continuity Clinics (NBSCCs). These clinics are integral to achieving optimal outcomes by providing follow-up oversight and assistance for individuals identified through screening. Continuity of NBSCC care for NBS during the COVID-19 pandemic was both challenging and necessary and was accomplished through innovative strategies of dedicated personnel. Following the discontinuation of the community quarantine, a situation assessment survey was completed by each NBSCC to better understand the challenges encountered and their effect on patient care. Performance data from each NBSCC were reviewed both before and after an extended community quarantine (2018–2021) to evaluate the impact of NBSCC disaster contingency plans in overcoming the resultant challenges (transportation, supply chain, etc.). Thematic analysis of the survey showed three primary challenges: Operations, communications, and safety. In 2018 and 2019, successful patient contacts were 70.6% and 70.2%, respectively. During the pandemic, successful contacts were 74.9% in 2020 and 76.8% in 2021, demonstrating that the contact approaches taken by the NBSCCs were sufficient to maintain (and even improve) patient contacts. The number of unresponsive patients decreased during the pandemic likely due to decreased mobility and improved follow-up actions from the NBSCCs.

## 1. Introduction

### 1.1. The Philippine Newborn Screening Program

Newborn bloodspot screening (NBS) was first introduced in the Philippines in 1996 by a research study group investigating the local incidence of several congenital disorders. Building on their incidence findings supporting the need for screening, the program slowly gained momentum. Support from the Department of Health (DOH), individual dedication and perseverance, and national law in 2004 all combined for a sustainable national program [1,2]. As the Philippine Newborn Screening Program (PNSP) grew, so too did its six-part support system (education, screening, short-term follow-up, diagnosis, management, and long-term follow-up/evaluation [3]. The last two of these components are included as roles of the PNSP’s Newborn Screening Continuity Clinics (NBSCCs) (note: See [4] for a detailed description of the establishment and responsibilities of the NBSCCs and [5] for a description of the elements and mechanisms for their performance evaluation). In short, the NBSCCs are focused on ensuring that the medical and other long-term follow-up needs of patients identified through newborn screening are met. Currently, there are 15 NBSCCs strategically located throughout the Philippines (Figure 1). The most recent continuity clinic based in Corazon Locsin Montelibano Memorial Regional Hospital in Bacolod City, Negros Occidental, was included in the network in January 2021. There are 17 administrative regions that primarily serve to coordinate planning and organize national government services across multiple local government units.

For over five years, the NBSCCs have ensured sustained and successful management of children diagnosed through the NBS. A detailed manual of operations exists and includes the follow-up flow (Figure 2), which outlines the steps for short- and long-term follow-up [6]. Through centralized case monitoring, newborns with positive newborn screening results (increased risk for a screened condition) are tracked and assisted through the medical care system to ensure good general pediatric care and anticipatory guidance. Care is taken to ensure that indigent patients are not overlooked. In cases where a patient fails to complete their follow-up process and cannot be satisfactorily tracked despite the best efforts of the NBSCC, DOH regional offices [Centers for Health Development (CHDs)] are contacted to provide additional follow-up assistance.

Despite the general successes of the NBSCC system in helping to ensure successful outcomes for PNSP-identified cases, several challenges to optimal long-term outcomes existed prior to the COVID-19 pandemic and continue today. Among these is the lack of disease specialists within certain regions, which is a major obstacle for conditions requiring specialized long-term disease management. This specialist deficiency is being addressed through (1) collaborations with specialists in other regions; (2) limited establishment of additional satellite clinics; and (3) telehealth patient visits. Furthermore, to help increase the number of physician specialists across the country, funding has been made available for physicians seeking fellowship training in pediatric endocrinology and clinical genetics [7]. A partnership with the Philippine Society for Pediatric Metabolism and Endocrinology was also established to develop additional follow-up clinics for routine care of patients with endocrine disorders, congenital hypothyroidism (CH), and congenital adrenal hyperplasia (CAH).

Archipelagic geography is also a challenge for the NBSCCs and for all service delivery organizations in the Philippines. When initially established, NBSCCs were intended to be located in every Philippine administrative region [8]. From the outset, because of geographic and related transportation challenges, there have been four NBSCCs that serve more than one administrative region (see Figure 1): (1) NBSCC-Philippine General Hospital (Manila); (2) NBSCC-Zamboanga City Medical Center (Zamboanga City); (3) NBSCC-Northern Mindanao Medical Center (Cagayan de Oro); and (4) NBSCC—Southern Philippines Medical Center (Davao City). Currently, the expanded availability of treatment is offered through additional satellite clinics in some regions. Additionally, a treatment network was established in Region VI (Western Visayas), Region XI (Davao), and Region XIII (Caraga) to provide additional long-term follow-up services. Currently, there are 19 satellite clinics across the islands of Panay, Negros Occidental, and Mindanao with plans to develop similar treatment networks in other regions. The aim continues to be the existence of at least one continuity clinic in every province in the Philippines.

Aside from the NBSCC and satellite clinics, clinical genetics centers also have been created in each of the three main island groups in the Philippines (Luzon, Visayas, and Mindanao) to respond to the acute and long-term needs of patients diagnosed through the PNSP. Currently, the services of the NBSCCs and satellite clinics are conducted on an outpatient basis and are limited to non-critical care services.

### 1.2. The COVID-19 Pandemic

The novel Coronavirus Disease 2019 (COVID-19) resulting from the SARS-CoV-2 virus has significantly affected all aspects of life worldwide since its discovery. Working to limit COVID-19 infections, quarantines and lockdowns have been common globally, and travel restrictions have existed both within and between countries. Recognizing the alarming spread and severity of infection, the World Health Organization (WHO) officially characterized COVID-19 as a pandemic on 11 March 2020 [8].

In the Philippines, the national government declared an “enhanced community quarantine” for Metro Manila beginning 15 March 2020 until 14 April 2020. Public transportation, non-essential businesses, and classes at all levels were suspended to restrict mobility and exposure. The entire population, except for essential workers, was placed under “stay at home” orders [9], later made more stringent by extending the quarantine to the entire island of Luzon [10]. In late June, Cebu City was identified as the epicenter of COVID-19 in the Philippines and a “hard lockdown” was initiated there [11]. In response to these limitations, the economy suffered markedly resulting in significant unemployment and extensive business closures.

Healthcare also was impacted. In response to the extreme increase in patients suffering from COVID-19, medical priorities were immediately shifted to respond to the health emergency. Outpatient services were reduced or discontinued, and elective procedures and surgeries were often forced to be deferred [12]. Public health services such as vaccination campaigns were postponed, increasing the risk of outbreaks of vaccine-preventable diseases in the Philippines. Unlike some countries where healthcare access for regular needs, including antenatal care and newborn screening, were reportedly unavailable, NBS continued [13].

Reported here are the results of a survey of NBSCCs that investigates the impact of COVID-19 on the services and, potentially, the health of newborns diagnosed and referred to NBSCCs during the pandemic.

## 2. Materials and Methods

Following the initiation of the community quarantine, a situation assessment survey to be completed online was distributed to the 14 NBSCCs in July 2020. An informed consent form was included in the distributed material and its completion was required prior to completing the survey. Survey questions concerned the impact of the pandemic on the NBSCC activities including challenges encountered and strategies implemented to address them. Survey findings were thematically analyzed using Braun and Clarke’s thematic analysis framework [14]. Data analyses were performed independently by four investigators from the National Institutes of Health’s Newborn Screening Reference Center. Following the analyses, NBSCC respondents and the Newborn Screening Reference Center analysts discussed the survey responses and reached a consensus as to appropriate and harmonized descriptions of the challenges encountered and the actions required to overcome them.

To compare NBSCC operations activities in pre-pandemic years (2018, 2019) and pandemic years (2020, 2021), patient interaction data from each NBSCC were obtained and analyzed. Data included annual and cumulative data for endorsed patients and were adjusted by excluding patients who died or were discharged after having been determined not to have the disorder (based on findings of an accredited specialist). It is the goal of the NBSCC to make contact with all endorsed patients at least once in a six-month period. Patients with whom contact is not made after numerous attempted contacts by the NBSCC and additional follow-up attempts by the relevant CHD are classified as “unresponsive”.

## 3. Results

There were 14 responses to the July 2020 survey, one from each NBSCC. The responses were analyzed and thematically categorized as follows: (1) Challenges to the NBSCC’s operational capacity; (2) challenges with patient communication; and (3) challenges with staff and patient safety.

### 3.1. Challenges to the NBSCC’s Operational Capacity

As a result of COVID-19, face-to-face clinic consultations were temporarily halted in 85.7% of the continuity clinics during the pandemic. The primary reason reported was the loss of manpower at the NBSCC due to the hosting institution’s decision to close its clinics. As a result of travel restrictions, 57.1% of NBSCCs also reported difficulties in distributing medical foods and medications. A reduction in the activities conducted by the continuity clinics such as subspecialty clinics and lay fora (patient education seminars) was reported by 28.6% of the NBSCCs.

### 3.2. Challenges with Patient Communications

Patient communications were a common challenge faced by the NBSCCs during the pandemic. Difficulty in patient recall was reported by 50.0% of the NBSCCs. In order to minimize patient contact, telemedicine consultations were initiated across all NBSCCs and quickly became an effective means of patient contact. Telemedicine and telegenetics visits were readily accepted as an alternative to face-to-face clinic visits, but these were not without difficulties [15]. Three continuity clinics (21.4%) reported difficulty conducting telemedicine consultations due to poor internet connections or lack of proper equipment to conduct consultations.

### 3.3. Challenges to Health Staff and Patient Safety

Only one continuity clinic (7.1%) reported difficulties with staff and patient safety during the pandemic. The remaining NBSCCs reported that any potential difficulties were overcome using telemedicine. Over a period of three months, the 14 continuity clinics saw a total of 379 patients via telemedicine consultations.

### 3.4. NBSCC Statistics

Annual and cumulative NBSCC data for 2018–2021 are presented in Table 1. A total of 6519 endorsed patients were under the supervision of the NBSCCs by the end of 2021. An increase in endorsements of 146% occurred between 2019 and 2020 corresponding to the 2019 expansion of NBS from 6 to 29 conditions [16,17]. The preponderance of the additional endorsements was from hemoglobinopathy and thalassemia cases. The decrease in the number of endorsed patients in 2021 corresponds to the decrease in national newborn coverage from 80.4% in 2020 to 70.7% in 2021 [18]. Throughout the four years, the number of patient contacts rose, keeping pace with the added endorsements. In 2018 and 2019, the contact percentages contacted were 70.6% and 70.2%, respectively, and for 2020 and 2021, these percentages increased to 74.9% and 76.8%. While management reorganization during this period likely contributed to the increased contact efficiency, it is clear that during the pandemic, patient contact services were not negatively affected and contact percentages improved slightly. The decrease in the percentages of unresponsive patients across the four years likely reflects both system improvements resulting from the management changes alluded to previously and decreased mobility of families because of the pandemic’s travel restrictions.

## 4. Discussion

Since the beginning of the pandemic, the NBSCCs have encountered new and unique issues and concerns affecting their operations. These in turn have affected the delivery of quality services to patients and their families. As a result of the imposed quarantine, modes of transport were limited, and transportation for both staff and patients was almost nonexistent. Strict border controls were enacted between provinces, challenging the way in which medical foods, supplies, and medicines could be transported. In some cases, medical items such as NBS specimens, which were transported to a provincial border, could be picked up by an ambulance on the other side of the border for transport to a medical facility. Patient movements were also possible in this way. Metabolic patients who required blood draws for monitoring also experienced difficulties in getting to NBSCCs or Newborn Screening Facilities (NSFs). For certain patients, such as those with endocrine disorders, CH, and CAH, monitoring laboratory services were only available at a limited number of facilities.

As government hospitals, 13 of the 14 host facilities of the NBSCCs were designated as COVID-19 Referral Centers by the DOH. Only one facility was not: West Visayas State University Medical Center (Iloilo) (see Figure 1). As referral centers, the organization of the host facilities was modified, and operations were streamlined to adequately respond to the pandemic. Of necessity, this included the temporary closure of outpatient services, including the NBSCCs.

In addition to the general categories of NBSCC challenges noted above, Table 2 summarizes some of the specifics of these challenges and the strategies employed to address them. Contingency plans were executed and there was coordination with the different stakeholders (i.e., Newborn Screening Centers (NSCs), Local Government Units, DOH-CHDs, Clinical Genetics Unit, and satellite clinics) to assure continued delivery of services to patients despite the limitations caused by the pandemic. NBSCCs utilized telemedicine consultations whenever possible as a substitute for face-to-face visits. The limitations of telemedicine visits were explained to the parents and caregivers prior to the visit. During the pandemic, five NBSCCs relied solely on telemedicine consultations while five others used both face-to-face and telemedicine consultations. In the latter case, telemedicine consultations were reserved for patients who did not need urgent or emergent medical care. All telemedicine encounters were well documented.

Despite having well-thought-out contingency plans for continuing operations in case of emergencies, some NBSCCs encountered difficulties in their implementation. For example, cellular phone calls, text messages, and telemedicine visits all rely on good Internet connections. Unfortunately, some areas of the country have poor Internet connectivity, which interfered with some transmissions. Additionally, some parents were not sufficiently familiar with the use of cell phones and the Internet, which slowed telemedicine services. For facilities that used face-to-face consultations, adequate clinic space was often an issue because of increased COVID-19 cases, the need for increased numbers of hospital beds, and related space requirements. Caregivers’ apprehensiveness about contracting the virus also resulted in a decrease in the number of patients attending clinics.

Coordination with other stakeholders to assure the delivery of services to NBSCC patients was a continuing consideration. Some NBSCCs requested assistance from the Local Government Units and DOH-CHDs to help in mobilizing healthcare workers (i.e., Barangay Health Workers, Rural Health Units, and health centers) to assist in caring for NBSCC patients. Care concerns included (1) the distribution of medications, medical formulas, and related supplies; (2) documentation of anthropometrics; (3) monitoring medical indicators for metabolic patients; and (4) assuring compliance with treatment.

## 5. Conclusions

The COVID-19 pandemic has provided the NBSCCs with opportunities to strengthen their ties with other NBS stakeholders and to think of creative ways to achieve their goals and objectives. The modified NBSCC contingency plans resulting from the challenges encountered during the COVID-19 pandemic can be used as references to better prepare for future disasters and/or pandemics. Given that the COVID-19 viral threat continues, along with the possibility of other unexpected events, regular and continuous evaluation and contingency planning are essential to continuing quality services for optimum patient care.

In conclusion, continuing care for patients under NBSCC oversight is essential for achieving optimum health outcomes. The COVID-19 pandemic resulted in new and innovative strategies for addressing challenges to NBSCCs’ operational capacity, mechanisms for patient communication, and adequate safety of staff and patients. Based on NBSCC data, patient services were continued at the same level during the pandemic as before and fewer patients were lost-to-follow-up. The work ethic and dedication of NBSCC staff members resulted in these continued high levels of service.

## Figures and Tables

**Figure 1 IJNS-09-00002-f001:**
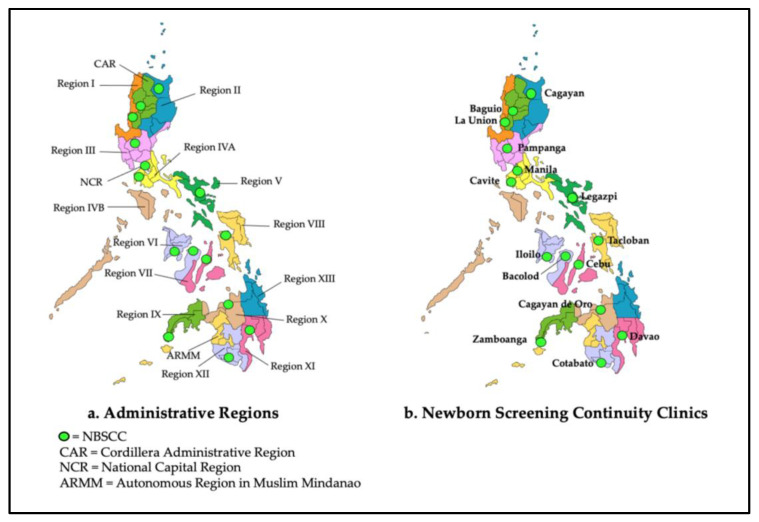
Administrative Regions of the Philippines and Newborn Screening Continuity Clinic Locations.

**Figure 2 IJNS-09-00002-f002:**
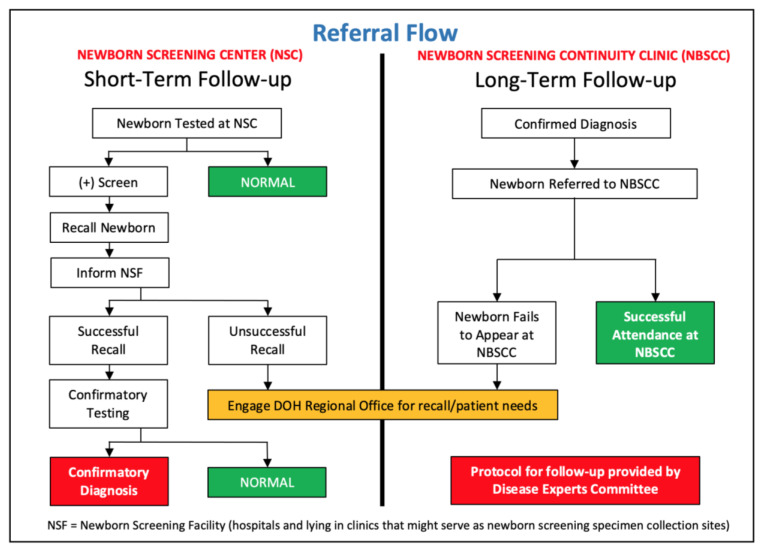
Short- and long-term follow-up flow for patients identified through newborn screening.

**Table 1 IJNS-09-00002-t001:** Annual and Cumulative Data from Newborn Screening Continuity Clinics 2018–2021.

Category	Year			
	2018	2019	2020	2021
Annual number of endorsed newborns (referred from NSCs)	459	476	1171	643
Cumulative endorsed newborns (long-term care monitoring)	4229	4705	5876	6519
Annual number of expired cases (died while in long-term care)	10	22	46	17
Annual number discharged (diagnosed clinically normal)	17	18	29	8
Actual newborn census [endorsed − (expired + discharged)]	3991	4427	5523	6141
Number contacted (contact with patient made within 6 months)	2819	3107	4138	4717
Percentage contacted [(number contacted/actual census) × 100%]	70.6%	70.2%	74.9%	76.8%
Annual unresponsive patients (patient contact lost for over 6 months)	304	148	65	39

**Table 2 IJNS-09-00002-t002:** Strategies initiated to address some of the challenges posed by the pandemic.

Challenges	Strategies
Limited or complete disruption of clinic hours	Guidelines from the Philippine Pediatric Society and University of the Philippines Manila were distributed defining the conduct of telemedicine consultations Collaboration with subspecialists was continued remotely through available telemedicine platforms The utilization of telemedicine consultations were maximized through use of online platforms, phone calls and SMS for continued patient follow-up.
Distribution of medical food, supplies, and medicine	Stakeholder coordination was strengthened including Local Government Units and the Department of Health—Centers of Health Developments’ abilities to assist the NBSCCs in the meeting patients’ medical needs.
Safety of both NBSCC staff and patients during face-to-face consultations	Personal protective equipment were provided for NBSCC staff members. Distancing and mask protocols were established for clinic visits and were strictly enforced.

## Data Availability

All data generated or analysed during this study are included in this published article.

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
