# Peer review of "Newborn Screening Long-Term Follow-Up Clinics (Continuity Clinics) in the Philippines during the COVID-19 Pandemic: Continuing Quality Patient Care"

_2409-515X, 2022, doi:10.3390/ijns9010002_

Round 1

Reviewer 1 Report

The manuscript titled “Newborn Screening Long-Term Follow-up Clinics (Continuity 2 Clinics) in the Philippines During the COVID-19 Pandemic: 3 Continuing Quality Patient Care” is a merely descriptive article of a local experience, describing interesting issues that newborn screening programs had to face during COVID-19 pandemic.

 It is a well-written article, however, previously to be published, I suggest to introduce some minor format editions:

 1. Abstract - Lines 46-47: I suggest to indicate the percentages of successful patient contacts with one decimal (the second ones do not have any statistical significance). This suggestion also applies to the percentages indicated in the Results section, Lines 205-206 and in Table 1.

 2. Introduction – Line 68: Please delete the word “and” in the sentence “Currently there and are 15 NBSCCs strategically located throughout the Philippines”.

 3. Results – Line 186: In order to unify the way to express the results along all the manuscript, please indicate the percentage of difficulty in patient recall with one decimal (50.0%).

 4. Results – Line 205: Please delete the additional space at the beginning of the sentence “In 2018 and 2019, the contact percentages…”.

5. In my opinion there are an overabundance of non-conventional abbreviations that are used only one or two times, that can be omitted in order to clarify the manuscript reading. For example, LGU, NSRC, NSC and NSF.

Author Response

REVIEWER 1

1.     Abstract - Lines 46-47: I suggest to indicate the percentages of successful patient contacts with one decimal (the second ones do not have any statistical significance). This suggestion also applies to the percentages indicated in the Results section, Lines 205-206 and in Table 1.

modified

...successful patient contacts were 70.6% and 70.2% respectively. During the pandemic, successful contacts were 74.9% in 2020 and 76.8% in 2021, demonstrating that the contact approaches taken

2.     Introduction – Line 68: Please delete the word “and” in the sentence “Currently there and are 15 NBSCCs strategically located throughout the Philippines”. 

"and" deleted in the sentence

3.     Results – Line 186: In order to unify the way to express the results along all the manuscript, please indicate the percentage of difficulty in patient recall with one decimal (50.0%).

 One decimal point added for this figure.

4.     Results – Line 205: Please delete the additional space at the beginning of the sentence “In 2018 and 2019, the contact percentages…”. 

Addressed – space deleted

5.     In my opinion there are an overabundance of non-conventional abbreviations that are used only one or two times, that can be omitted in order to clarify the manuscript reading. For example, LGU, NSRC, NSC and NSF. 

The acronyms LGU and NSRC were removed. However, the acronyms NSC and NSF were maintained as they also appeared in Figure 2.

Reviewer 2 Report

This a well-presented account of the impact of COVID-19 on NBS follow-up care in the Philippines. It is impressive that patient contacts were maintained over 2020-21, largely as a result of a pivot to tele-health. 

Minor comments only.

1. Fig 2. please explain the acronym NSF

2. Is it possible to include the survey as an additional material? It is also not clear how many responses were received and whether these included all of the NBSCCs.

3. The increase in endorsed patients from 2019 to 2020 is explained by the addition of haemoglobinopathy cases but the decrease in 2021 is not explained. Do the authors have data on the the total number of babies screened and is the drop likely to be explained by decreased screening coverage? Or were protocols modified so that less babies were referred?

4. The duration of follow-up is unclear but may be lifelong for some conditions.  Is it possible to split % contacted by disorder or patient age?  For example, it may be that % contacts were maintained due to an increase in young patients and higher % contact within this group. 

Author Response

REVIEWER 2

1.     Fig 2. please explain the acronym NSF

- A note was added at the bottom of the Figure to explain the acronym NSF

NSF – Newborn screening facility (hospitals and lying-in clinics that might serve as newborn screening specimen collection sites)

2.     Is it possible to include the survey as an additional material? It is also not clear how many responses were received and whether these included all of the NBSCCs. 

Line 173-174:

There were 14 responses to the July 2020 survey, one  received from each NBSCC.. The responses were analyzed...

3.     The increase in endorsed patients from 2019 to 2020 is explained by the addition of haemoglobinopathy cases but the decrease in 2021 is not explained. Do the authors have data on the total number of babies screened and is the drop likely to be explained by decreased screening coverage? Or were protocols modified so that less babies were referred? 

Line 204-206: The decrease in the number of endorsed patients in 2021 corresponds to the decrease in the national newborn screening coverage from 80.4% in 2020 to 70.7% in 2021 [18]. 

4.     The duration of follow-up is unclear but may be lifelong for some conditions.  Is it possible to split % contacted by disorder or patient age?  For example, it may be that % contacts were maintained due to an increase in young patients and higher % contact within this group. 

We inserted ’lifelong’  in line 284 in an effort to clarify the intent of NBSCC follow up.

Unfortunately, we do not have the split% contacted by disorder or patient age.

We thank the reviewer for this comment and will  consider it in future planning.
